# Vitamin D Deficiency Does Not Affect Cognition and Neurogenesis in Adult C57Bl/6 Mice

**DOI:** 10.3390/nu16172938

**Published:** 2024-09-02

**Authors:** Mark Doumit, Carla El-Mallah, Alaa El-Makkawi, Omar Obeid, Firas Kobeissy, Hala Darwish, Wassim Abou-Kheir

**Affiliations:** 1Department of Anatomy, Cell Biology, and Physiology, Faculty of Medicine, American University of Beirut, Beirut P.O. Box 11-0236, Lebanon; mnd09@mail.aub.edu (M.D.);; 2Department of Nutrition and Food Science, Faculty of Agriculture and Food Sciences, American University of Beirut, Beirut P.O. Box 11-0236, Lebanon; 3Department of Biochemistry and Molecular Genetics, Faculty of Medicine, American University of Beirut, Beirut P.O. Box 11-0236, Lebanon; 4Hariri School of Nursing, American University of Beirut, Beirut P.O. Box 11-0236, Lebanon

**Keywords:** vitamin D, deficiency, supplementation, microglia, neural stem cells, neurogenesis, astrocytes, sex differences

## Abstract

Vitamin D deficiency is a global problem. Vitamin D, the vitamin D receptor, and its enzymes are found throughout neuronal, ependymal, and glial cells in the brain and are implicated in certain processes and mechanisms in the brain. To investigate the processes affected by vitamin D deficiency in adults, we studied vitamin D deficient, control, and supplemented diets over 6 weeks in male and female C57Bl/6 mice. The effect of the vitamin D diets on proliferation in the neurogenic niches, changes in glial cells, as well as on memory, locomotion, and anxiety-like behavior, was investigated. Six weeks on a deficient diet was adequate time to reach deficiency. However, vitamin D deficiency and supplementation did not affect proliferation, neurogenesis, or astrocyte changes, and this was reflected on behavioral measures. Supplementation only affected microglia in the dentate gyrus of female mice. Indicating that vitamin D deficiency and supplementation do not affect these processes over a 6-week period.

## 1. Introduction

The vitamin D receptor (VDR), as well as its metabolites and enzymes, are highly distributed in the brain, which suggests a role in brain function [1]. Vitamin D is a micronutrient with a secosteroid structure that is known for its role in various physiological mechanisms such as bone mineralization, kidney function, immune response, calcium homeostasis, and proliferation and differentiation [2]. While it is mostly derived from exposure to the sun, vitamin D deficiency remains prevalent among some populations [3,4,5,6] and is associated with multiple diseases ranging from breast cancer to insulin-dependent diabetes [7,8]. Vitamin D deficiency has also been implicated to play a role in brain-related diseases and disorders such as Parkinson’s disease (PD) [9,10,11,12], schizophrenia (SCZ) [13,14,15], depression [16,17,18], Alzheimer’s disease (AD) [19,20,21], and multiple sclerosis (MS) [22,23,24]. More recent evidence suggests that brain and cerebrospinal fluid 25(OH)D is associated with lower odds of dementia and mild cognitive impairment [25,26]. There have also been cases of disorders like autism [27,28] and trichotillomania where people have benefitted from vitamin D therapy [29,30].

Some estimate a billion people worldwide have vitamin D deficiency [5]; thus, it is imperative that we understand how vitamin D deficiency can contribute to the etiologies and progression of neuropsychiatric and neurodegenerative diseases [31]. The evidence relating vitamin D and these diseases is mixed. Meta-analyses and systematic reviews have revealed either weak or no associations with significant methodological variances and discrepancies between research protocols and measures [31,32,33,34]. This complexity can also be attributed to the intricacies of the vitamin D system. Deciphering the genomic, non-genomic, and epigenetic effects it exerts and understanding how they interact across neurons, glial, immune, endothelial, and ependymal cells pose significant methodological challenges.

Vitamin D has been found to be neuroprotective by modulating nerve growth [35,36] and decreasing L-type calcium channel expression [37]. It is involved in regulating reactive oxygen species [38,39], inducing neurotrophic factors [35,40,41,42,43], and nitric oxide synthase [38]. Furthermore, vitamin D and its metabolites are involved in other neuroprotective mechanisms, including amyloid phagocytosis and clearance [44,45,46], decrease of alpha-synuclein aggregation [47], as well as anti-inflammatory [48,49,50] and anti-oxidative processes [51,52].

However, the evidence regarding vitamin D’s effects on the brain has been conflicting, with deficiencies and supplementation showing different outcomes depending on the species and strain of rodents. In vitro and in vivo studies looking into the vitamin D system have found evidence to suggest that it plays many roles in the central nervous system (CNS), with different cell types responding differently to either increased or decreased vitamin D or its receptors [33,34,53,54,55,56,57]. Vitamin D’s effect in vitro on neural stem cells has also been complex [58,59,60]. This complexity underscores the challenges of understanding and harnessing the potential benefits of vitamin D. Nonetheless, manipulating vitamin D levels has been associated with cellular and molecular changes [61,62,63,64,65,66,67,68,69,70,71,72].

Therefore, we sought to investigate the effect of vitamin D supplementation and deficiency on proliferation and neurogenesis in the main neurogenic niches of the brain: the dentate gyrus (DG), which is also necessary for episodic learning and memory [73], and the subventricular (subependymal) zone (SVZ). In addition, we explored the effect of diet on microglia and astrocyte morphology in the DG and conducted experiments to assess memory, locomotion, and anxiety-like behavior.

## 2. Materials and Methods

### 2.1. Design and Protocol

#### 2.1.1. Animals and Study Design

Experiments were performed on 6–8 month-old male and female C57BL/6 mice in accordance with the National Institutes of Health Guidelines for Animal Research (Guide for the Care and Use of Laboratory Animals) [74], the 3Rs principles [75,76], and under a protocol approved by the Institutional Animal Care and Use Committee (IACUC) at the American University of Beirut (AUB) (Approval #17-05-414) [77]. Animals were maintained in a controlled environment, at a temperature of 20–22 °C, and under a 12 h reverse light/dark cycle. Animals were provided with water and food ad libitum. C57BL/6 male and female mice were bred at the Animal Care Facility, Faculty of Medicine, AUB. To habituate the mice to the experimenter who performed the behavioral testing, they were handled at least 3 times a week at 2 months before the start of the protocol. Mice were also provided in-cage shelters and additional bedding material, such as autoclaved tissue or cardboard. Litters were counterbalanced across groups to decrease any specific effects related to any particular pregnancy or dam. Male and female C57BL/6 mice were divided into 3 groups: one group received 0.1 times the recommended daily intake (RDI) of vitamin D3 and was the vitamin D deficient group (VDDG), a group received 1× RDI and was the control group (VDCG), and a group received 2.4× RDI and was the vitamin D supplemented group (VDSG).

#### 2.1.2. Diet

The diet was made in-house by the Department of Nutrition, Faculty of Agriculture and Food Sciences, at the American University of Beirut, Lebanon. A vitamin D-free AIN vitamin mix (Vitamin D Deficient AIN-93-VX Vitamin Mix-DYET# 319255) was added into a mixer with casein (vitamin-free), L-methionine, corn starch, sucrose, oil, cellulose, mineral mix, choline bitartrate, and a variable amount of vitamin D3 depending on the group, i.e., 100 IU/kg for the VDDG, 1000 IU/kg for the VDCG, and 2400 IU/kg of diet for the VDSG. The composition of the diets can be found in Appendix A.

#### 2.1.3. BrdU Administration

5-Bromo-2′-deoxyuridine (BrdU), a thymidine analog, is a proliferation marker for stem/progenitor cells because it incorporates into DNA during a cell’s S-phase. Consequently, when BrdU is administered to mice, it substitutes thymidine nucleotides and can be detected by binding to a fluorescence-conjugated antibody. Mice were injected with BrdU (Sigma-Aldrich, B5002-1G, St. Louis, MO, USA, 66 mg/kg/i.p. injection of 0.1 mL) dissolved in 0.9% warm sterile saline. The injections were given 8 and 9 days before they were euthanized (1 injection per day). This resulted in 2 injections per mouse, totaling 132 mg/kg BrdU per mouse [78].

#### 2.1.4. Tissue Preparation

First, the mice were anesthetized using a xylazine/ketamine mix for induction and isoflurane for maintenance [79,80]. After gradual onset of anesthesia, the animals were checked for a righting reflex after they became immobile; a toe pinch was used to further assess depth of anesthesia, i.e., any signs of pain or sensation such as withdrawal or vocalizations [79,80]. Blood was taken from the left ventricle of the heart with a syringe and 31-gauge needle while the mice were maintained on isoflurane. The blood was centrifuged and the serum placed in microcentrifuge tubes and frozen at −20 for later analysis with a Roche Cobas 8000 (e801)—an electrochemiluminescence binding assay (Roche Diagnostics GmbH, Mannheim, Germany). The details and results can be found in Appendix A (results presented in Appendix A).

After blood withdrawal, the mice underwent cardiac perfusion with 4% paraformaldehyde (PFA) while being maintained on isoflurane. Brains were removed and fixed overnight in 4% PFA. After which, they were placed in 20% sucrose and then stored in a 30% sucrose solution for cryoprotection. According to the fractionator principle, systematic random sampling of brain sections was performed [81]. In brief, 40 μm coronal frozen sections were serially cut using a freezing microtome from the rostral SVZ to the caudal DG. They were cut at coordinates covering almost the entirety of the SVZ and hippocampal formations [82].

The SVZ distribution of areas was as follows: anterior distribution from 1.18 to 0.74 mm relative to the bregma, intermediate distribution from 0.74 to −0.14 mm, posterior distribution from −0.14 to −0.94 mm, post-posterior distribution ranging from −0.94 mm to −2.1 mm [83]. The DG region was divided into three areas as follows: rostral distribution ranging from −0.94 to −2.1 mm relative to bregma, intermediate ranging from −2.1 to −2.9, and caudal ranging from −2.9 to −4.04 [82]. The post-posterior SVZ and rostral DG shared the same well-plate row (−0.94 to −2.1 mm relative to bregma).

Each SVZ region was collected in eight sets containing an average of 1.375 rostral, 2.75 intermediate, 2.5 posterior, and 3.62 post-posterior slices, with each slice being 280 µm apart from the next. Each DG region was collected in eight sets containing an average of 3.62 rostral, 2.5 intermediate, and 3.56 caudal slices, with each slice being 280 µm apart from the next (Figure 1). Sections were placed in 0.1 M phosphate-buffered saline (PBS) containing 15 mM sodium azide and stored at 4 °C for future processing. Sections from the intermediate SVZ and caudal DG were used to assess proliferation and differentiation, as these areas have been noted as having the greatest numbers of proliferating cells [83,84,85] and a significant role in contextual memory formation and retrieval [73].

#### 2.1.5. Immunofluorescence

For BrdU detection, free-floating sections of the mice brains were incubated in 2 N HCl for half an hour at 37 °C. Sections were washed 3 times with 0.1 M PBS, and the acidic effect was neutralized with 0.1 M Sodium Borate (pH 8.5) for 10 min at room temperature. Brain sections were washed with 0.1 M PBS and transferred to the blocking and permeabilization solution (10% FBS, 0.3% Triton-X diluted in PBS) for 1.5 h at 4 °C. Sections were incubated overnight at 4 °C with rat monoclonal anti-BrdU (1:500; ab6326, Abcam, Cambridge, UK) and rabbit anti-NeuN (1:500; ab104225, Abcam, Cambridge, UK) antibodies. The next day, sections were washed with PBS, and the secondary antibodies Alexa Fluor-594 anti-rat (1:500; ab150160, Abcam, Cambridge, UK) and Alexa Fluor-488 anti-rabbit (1:500; ab150077, Abcam, Cambridge, UK), diluted in PBS with 1% FBS and 0.3% Triton-X, were added for 2 hrs. Finally, sections were mounted onto slides with Fluoromount-G (Thermo Fisher Scientific, Hanover Park, IL, USA) ™ or Fluoro-Gel containing DAPI (Electron Microscopy Sciences, Hatfield, PA, USA) and covered with a thin glass coverslip.

Apart from incubating sections in HCl and sodium borate, a similar protocol was used for the rest of the immunostaining. For the detection of GFAP (1:500, astrocyte marker, Encor-MCA-5C10, Gainesville, FL, USA) and IBA1 (1:500, microglial marker, Encor-RCPA-IBA1, Gainesville, FL, USA) staining, free-floating sections were rinsed in 0.1 M PBS and incubated overnight with the primary antibodies. Sections were then incubated in the secondary antibodies, Alexa Fluor-488 anti-rabbit (1:500; ab150077, Abcam, Cambridge, UK) and Alexa Fluor-568 anti-mouse (1:500; ab175473, Abcam, Cambridge, UK), for 2 h in the dark at room temperature. Sections were then mounted and left to dry before imaging.

#### 2.1.6. Imaging and Quantification

Quantitative analysis for the changes in adult neurogenesis in response to vitamin D3 deficiency and supplementation was assessed using fluorescent microscopy of BrdU-positive cells in all three groups. One set of slices was chosen randomly, and BrdU+ cells were counted in every 8th section (280 μm apart) using the 40× objective. Cell stereology was confined to the GZ and SGZ of the DG and the SVZ of the lateral ventricles. The total observed number of BrdU-positive cells across all sections was divided by the number of sections to normalize the number for comparison.

The microscopic analysis was performed using a Zeiss LSM 710 laser scanning confocal microscope and a Leica DM6 B Upright Microscope (Leica DFC7000 T camera) (Leica Microsystems Private Limited Company, Wetzlar, Germany). Brdu+ and BrdU+/NeuN+ cells were counted at 40×. GFAP and Iba1 Z-stack images were captured at 20×, then processed and analyzed using FIJI (ImageJ, version 1.54f) [86,87]. Appendix A provides the steps and codes for processing and analyzing microglia and astrocyte images. Appendix A includes 50×-oil objective images further detailing the process.

### 2.2. Behavioral Tests

#### 2.2.1. Open Field Test

The open field test (OFT) assesses anxiety-like behavior and locomotion. We used four rectangular open fields (38 cm wide × 38 cm long × 38 cm high) made of gray plastic in a quiet room under diffuse fluorescent lighting. The four open fields allowed for the testing of four mice simultaneously. A camera was suspended above the apparatus to record the trials and further analyze the results. Through the tracking software (AnyMaze v7.41), each box was divided into central and peripheral areas (9 cm from the wall), the comparison of which would suggest preference for area over the 6-min trial. A larger time spent in the periphery over central areas would demonstrate less anxiety-like behavior. All behavioral tests were performed at the end of the 6-week protocol. The OFT was performed 2 days before euthanasia.

#### 2.2.2. Temporal Order Recognition Test (TOR)

Innately, rodents spend more time exploring novel objects and objects in novel locations than familiar objects and locations. While spontaneous and novel object (or spatial) recognition tests are usually used to assess hippocampal-related memory, the temporal order recognition test (TOR), the ability to discriminate novelty based on recency, develops at a later age (by days) [88] and is dependent upon functional connectivity between more brain regions [89,90].

The four rectangular open fields used for the open field test were used for the TOR. Objects made from plastic building blocks were used for testing. The objects were similar in material and size (4–7 cm × 4–7 cm × 4–7 cm), but distinctively different in shape and color. The objects were secured to the floor of the open fields with double-sided tape [91].

Each mouse was tested in a series of 3 four-minute trials with an inter-trial interval (ITI) of approximately 1.5 h. Mice were placed in the center of the open field at the beginning of each trial and allowed to freely explore. At the end of each trial, mice were removed from the open fields and placed in their home cages next to each testing area for the inter-trial time. The open fields and objects were cleaned with 70% ethanol during the intertrial intervals. During Trial 1, the open fields contained two identical objects on one side of the box. After the 1st ITI, during Trial 2, two identical novel objects were introduced in the same area as the previous objects in each open field. For the test trial, or Trial 3, an exact copy of the object from Trial 1 and an exact copy of the object from Trial 2 were placed in the open box in the same positions as in previous trials. Response preference to the older or newer object should test temporal order recognition, where mice are expected to spend more time on the less recent object. To assess this preference, we calculated a relative discrimination index (DI) based on the ratio of time spent exploring the novel object: (the time exploring the older (more novel) object—the time spent exploring the newer (less novel) object) divided by total exploration time [92]. The TOR was performed on the day before the euthanasia.

### 2.3. Statistical Analyses

Data was analyzed using statistical software packages IBM SPSS Statistics v26.0 and GraphPad Prism v9.0. One-way ANOVAs were used to analyze the measures of interest across vitamin D groups. The normality of residuals and heteroscedasticity was assessed, and Kruskal–Wallis or Brown–Forsythe tests were performed if the assumptions were not met. Tukey corrections were used for post-hoc multiple comparisons of ANOVA tests, while Dunnett’s T3 was used for non-parametric ANOVAs. Summary statistics are presented as mean ± standard error of the mean.

## 3. Results

### 3.1. Vitamin D Deficiency Did Not Alter the Weight of Both Male and Female Mice

A two-way repeated measure ANOVA was used to assess the effect of time, diet, and the interaction of time × diet on final weight and weight change from the start to the end of the protocol. There was no significant effect of diet, but time × diet for the female mice was significant despite no significant differences between groups in the post-hoc multiple comparisons (Figure 2).

### 3.2. Vitamin D Deficiency and Supplementation Had No Effect on Motor, Cognitive, or Affective Functions in Male or Female Mice

#### 3.2.1. Vitamin D Did Not Affect Anxiety-like Behavior or Locomotion in Male or Female C57Bl/6 Mice

The OFT was used to assess locomotion and anxiety-like behavior. Videos were captured and analyzed later using video tracking software. Measures of interest included a preference index (PI) (i.e., the time spent in the periphery divided by the total time), distance traveled, and speed. One-way ANOVAs did not reveal any significant differences on measures of locomotion (speed or distance travelled). There were no significant differences between conditions on distance traveled in the periphery or central areas, or on the PI in either sex, indicating no effect on anxiety-like behavior either (*p* > 0.05) (Figure 3). It is worth noting that the estrous cycle was tracked through vaginal lavage and did not have a confounding effect on behavioral tests [93].

#### 3.2.2. Vitamin D Did Not Affect Hippocampal-Dependent Memory on the TOR in Male or Female C57Bl/6 Mice

The TOR was used to assess context-dependent memory within the span of approximately 3 h. We assumed the increased sensitivity of this test would be able to assess the small effect size vitamin D deficiency might lead to. A relative discrimination index (DI) was calculated based on the ratio of time spent exploring the novel object: the time exploring the older (more novel) object minus the time spent exploring the newer (less novel) object divided by total exploration time [92]. One-way ANOVAs did not yield any significant differences when comparing between vitamin D groups of either sex on measures of relative DI, speed, or distance traveled in the probe trial (Figure 4).

### 3.3. Vitamin D Deficiency Did Not Alter Neural Stem Cell Proliferation and Differentiation

BrdU+ cells in the granular and subgranular zones of the DG and around the SVZ were counted manually using a Leica DM6 B Upright Microscope (Leica Microsystems Private Limited Company, Wetzlar, Germany) and a tap counter. One-way ANOVAs were used to test whether there were significant differences between the vitamin D groups. There were no significant differences between the vitamin D groups, in the DG or SVZ, across both sexes, indicating no effect of vitamin D on proliferating cells in the neurogenic niches of the brain (Figure 5, Figure 6, Figure 7 and Figure 8, A & B).

BrdU+/NeuN+ double positives were counted manually using a Leica DMB6 upright fluorescence microscope, simultaneous Blue–Green–Red fluorescence through the eyepiece, and a tap counter. One-way ANOVAs were used to test whether there were significant differences between the vitamin D groups. There were no significant differences between the vitamin D groups, in the DG or SVZ, across both sexes (Figure 5, Figure 6, Figure 7 and Figure 8, A & C).

### 3.4. Vitamin D Affected Microglial Morphology in the DG of Female Mice Only, but Did Not Affect Astrocytes

We used anti-Iba1 (microglia) and anti-GFAP (astrocytes) antibodies to assess whether vitamin D manipulation affected the behavior of microglia and astrocytes. We assessed microglia and astrocytes on six measures: The *Perimeter* feature measures the perimeter of the cells and processes within the selected region of interest (ROI; the DG); *Fluorescence Intensity* (thresholded integrated density) refers to the sum of the thresholded pixels in the selection; the *Average Size* measure refers to the average size of microglial cells and processes; and *Circularity* is defined as the roundness of the extracted features of the cells and processes. “A value of 1.0 indicates a perfect circle. As the value approaches 0.0, it indicates an increasingly elongated shape” [94], and *Fluorescence in DG* represents the average area occupied by the cells within the ROI.

Microglial morphology in male mice did not change as a result of diet (Figure 9), neither did astrocyte count and morphology in male. However, the microglial morphology and count were altered significantly in female mice (Figure 10); the VDSG had a statistically significant lower fluorescence intensity, perimeter, and average size than VDCG (*p* < 0.05) and a significantly higher count compared to both VDCG (*p* < 0.01) and VDDG groups (*p* < 0.05) (post-hoc multiple comparisons were performed using Tukey for ANOVAs and Dunnett’s T3 for significant Brown–Forsythe ANOVAs on GraphPad Prism 9.0). However, no significant microglial differences were observed between the different diets in the male group (*p* > 0.05). Moreover, there were no significant differences in astrocytic measures of circularity, perimeter, integrated density (thresholded fluorescence intensity), average size, percentage of Region of Interest (DG) covered, or an automated count for male (Figure 11) or female (Figure 12) mice on different diets (*p* > 0.05).

## 4. Discussion

This study investigated the role of vitamin D on C57Bl/6 adult male and female mice to shed light on the possible roles of vitamin D deficiency and supplementation on proliferation in the neurogenic niches (DG and SVZ), microglial and astrocytic morphology in the DG, as well as motor, cognitive, or affective measures. By manipulating the amount of vitamin D in the rodent diet, we were able to model adult vitamin D deficiency and supplementation. Adult vitamin D deficiency (AVD) and developmental vitamin D deficiency (DVD) studies usually provide mice or rats with a modified diet that has lower than recommended vitamin D concentrations (1 IU/g).

Here, we used a 6-week protocol on mice aged 6–8 months, as opposed to previous studies, which employed longer (10 weeks to 14 months) deficiency and/or supplementation periods and younger mice. Although our 6-week protocol induced deficiency, it is possible that longer protocols were more likely to find cellular and behavioral changes. However, this too has been inconsistent [61,62,65,67]. For example, Liang, Cai, Duan, Hu, Hua, Jiang, Zhang, Xu and Gao [70] found significant differences between mice given postnatal vitamin D at “overdose” levels and the deficient and control groups at 6 weeks but not at 17 weeks. Thus, it indicates that the effect of vitamin D is non-linear and time-dependent, with possible compensatory feedback mechanisms.

The 2.4× RDI diet used in this study was based on manufacturer concentrations of vitamin D found in regular rodent diets and based on previous investigations into the average concentration of laboratory chow [95]. Moreover, placing mice on vitamin D-deficient diets *under* 0.25× RDI has been shown to lead to deficient levels of vitamin D after 4 to 6 weeks while avoiding hypocalcemia and elevated cut-offs for PTH levels [95,96]. The vitamin D-deficient diet we provided contained 0.1× RDI, as the complete absence of vitamin D does not reflect real-world conditions.

### 4.1. 25(OH)D Serum Levels Indicate Deficiency According to the EFSA Guidelines

Vitamin D deficiency was achieved in the VDDG, according to guidelines by the Endocrine Society (2011) and the EFSA (2016), indicating that deficiency can be reached within 6 weeks in adult C57Bl/6 mice. This corroborates previous studies that have found that less than 6 weeks is enough time to reach deficiency in younger female C57Bl/6 [96] and male and female FVB mice [95]. However, 6-week deficiency and supplementation did not have a significant effect on most behavioral or cellular measures. This can be due to the short duration of deficiency, 2–3 weeks to achieve deficient levels, 2–3 weeks being deficient, and the younger age of the mice.

A limitation of our study is that we used the Roche electrochemiluminescence assay. While HPLC-MS/MS is the preferred method for measuring any vitamin D metabolite, we performed ECLIA as a less-expensive alternative to HPLC-MS/MS to confirm the diet placed the mice within the expected range of vitamin D. Moreover, the Roche is an assay designed for human and clinical use, with a larger coefficient of variation than other tests such as radioimmunoassays and chemiluminescent assays [97,98]. However, taking into account the number of previous studies that have confirmed that 3 to 4 weeks is sufficient time to decrease circulating 25(OH)D to deficiency levels in mice [95,96], and 6 weeks is enough for brain 25(OH)D levels (in rats at least) to significantly change compared to those receiving a control diet (1000 IU/kg) [99].

These cut-offs are also subject to at least some debate and, to a certain extent, depend on population-level, epidemiological, or meta-analytic studies looking at relative risks for bone density, osteoporosis, and related skeletal problems (e.g., risk of hip fracture) [100,101]. Despite recent meta-analyses showing no effect of vitamin D level on fractures and bone densities in older adults in the general population [102,103,104], other studies did find some decreased risk for general or extra-skeletal outcomes such as mortality and/or cancer [104,105,106]. Moreover, vitamin D supplementation has been shown to be effective for some outcomes and/or for subpopulations with deficiency or certain diseases [107]. However, these meta-analyses suffer from the limitation of pooling all subjects into one or more stratified categories that do not usually account for the differences in absorption and/or metabolism due to genetic or individual differences. Nonetheless, general recommendations don’t seem to be guided by studies relating to mortality and morbidity and are not informed by the possible effect of deficiency on brain processes.

### 4.2. No Effect of Vitamin D on Proliferation, Differentiation, or Behavioral Measures

Vitamin D supplementation has been shown to increase proliferation, neurogenesis, and/or oligodendrogenesis in a model of Alzheimer’s disease in mice [71], a model of EAE/MS in rats [108,109], and cell cultures of neural stem cells exposed to neurotoxic or demyelinating agents or misaggregating proteins [59,60]. Whether these effects are due to decreased apoptosis [44,110,111,112], or increased proliferation or differentiation [36,59,60,108,113,114,115,116], and through which mechanisms is still inconclusive [113].

Although 2-hit models have provided evidence to support vitamin D’s beneficial effects, AVD models have struggled to find a detrimental effect of deficiency or a beneficial effect of supplementation. DVD and AVD studies in rats have been scarcer than in mice, but studies in both species have shown inconsistent results due to vitamin D manipulation. Evidence supporting behavioral differences between deficient, control, or supplemented rodents has been sparse, with only some tests picking up on differences related to memory, attention, or hyperlocomotion. And when found, these effects seem to be strain-dependent [61,63,67,68,70,72] and/or sex-dependent [66]. Other studies have found no differences between deficient and control or supplemented rodents [62,64,69].

In this study, vitamin D deficiency and supplementation did not affect cognitive function as assessed with the OFT and TOR. The OFT measures locomotion and anxiety-like behavior, while the TOR is a more sensitive version of the object recognition test, which assesses hippocampal-dependent memory. The effects of vitamin D on locomotion and memory have been difficult to interpret. For example, in mouse AVD models, [67] found significant differences on the OFT in 10-week-old C57Bl/6 and BALB/c mice kept on a vitamin D-deficient diet for 10 weeks. On the other hand, [62] did not find an effect of vitamin D on the OFT or the object recognition test in 22–23-month-old mice on a diet for more than 12 months. [117] also did not find any effect of vitamin D supplementation for 4 weeks on the OFT or EPM (which measures anxiety-like behavior). In the study most similar to ours, perhaps [118] et al. found an effect of deficiency on 8-month-old C57Bl/6 mice placed on a deficient diet for 8 weeks on the object recognition test. We did not find an effect of 6 weeks on a vitamin D-deficient or supplemented diet in 6–8-month-old male or female mice on these measures. The reasons for such differing results could simply be due to the age, strain, and time on a deficient diet.

However, we believe this to be too simplistic of an explanation considering there have been other results that have contradicted each other with very similar age, strain, and time on diet, for example on GAD65/67, 5-HIAA, HVA, or 5-Hydroxytryptamine protein levels [67,69]. The vitamin D system is complex; it includes two receptors (VDR and PDIA3) that are expressed across glial, neuronal, and ependymal cells [1], and this action can be epigenetic, genetic, and/or non-genomic [34,117,119]. VDR also interacts with steroid hormones [120,121,122], and it has epigenetic effects that can be passed transgenerationally through hormonal imprinting [123,124]. Therefore, we consider treating this system as a nonlinear system, where small changes in starting conditions can lead to large differences in outcome, a more reasonable approach towards the future interpretation of the effects of vitamin D in the brain (another highly complex system). Overall, despite significant differences found in some studies, on some cognitive and behavioral measures, in some strains, and on some tests [61,67,125,126], the weight of the evidence supports no to very minimal effects of low adult vitamin D on behavioral performance in C57Bl/6 adult mice.

Cellular proliferation and neurogenesis comparisons have also been mixed, with some finding a significant difference between deficiency and supplementation [71] and others not finding any between deficient and control mice [65]. The effect on cellular activity has also been mixed, with some groups finding differences in multiple neurotransmitter systems [67,69,123], and on measures of proliferation [59], neurogenesis [71], and differentiation [108,127]. Due to this *seemingly* contradictory evidence, perhaps the paradigm with which we investigate adult vitamin D deficiency and supplementation needs to be updated.

The lack of clear, robust behavioral and/or cognitive differences is echoed by the results for hippocampal and subventricular proliferation and neurogenesis. It is possible that some transcriptional changes might have begun [128] and may be specific to a certain neurogenic, oligodendrogenic, or astrogenic (proliferative) niche and therefore take a longer time to translate into cellular change [129]. This would reconcile some of the mixed results from in vivo studies with each other and with in vitro studies that provide evidence of a change in proliferation and/or differentiation into neurons, astrocytes, or oligodendrocytes. It might even elucidate some of the mechanisms and processes of radial glial or neural stem cell lineage, genealogy, and cell fate [130,131,132], a still ongoing debate that in turn sheds light on the effect of vitamin D.

Overall, what these studies inform us is that there is a need to have an adequate range of amino acids and macronutrients to enable normal physiological processes, at least on a population level. Any “inadequacy” can lead to aberrant processes, and an inadequacy in a nutrient that is involved in many diverse transcriptional responsibilities as well as epigenetic and non-genomic effects could possibly lead to a pathophysiological etiology and/or exacerbate it.

### 4.3. Sex-Specific Mircoglial Differences Due to Vitamin D Manipulation

The effect of vitamin D on astrocytes and microglia has not been a topic of interest despite studies showing that vitamin D receptors and enzymes are expressed in these cells [1]. Moreover, glial cells (microglia, astrocytes, and oligodendrocytes), neurons, and the peri-neuronal nets in the extra-cellular matrix form the basis of the tetrapartite synapse [133], which plays a pivotal role in learning and memory. Microglia and astrocytes both play a role in cognitive processes. Microglia, for example, play a role in synaptic pruning, and a decrease in their normal processes can affect this role and lead to decreased synaptic transmission, functional connectivity, and social behaviors [134]. While astrocytes have been implicated in memory and learning processes through their release of gliotransmitters, calcium activity, and metabolic coupling with neurons [135], vitamin D has also been shown to affect the proliferation, morphology, and maybe more importantly, the phenotype of both astrocytes [59,136,137,138] (including radial glial (stem) cells and mature astrocytes) and microglia [38,50,108,136,139,140,141,142].

There is a wealth of information about the effect of vitamin D on immune cells [122,141,142,143], and recent evidence suggests that microglia and astrocytes are affected more, across more areas of the hippocampus, by vitamin D deficiency and supplementation than neuronal cells [119], and that microglia might even play a role in stem cell differentiation [108,144].This evidence seemingly indicates that vitamin D affects the cellular proliferation of microglia and astrocytes, perhaps more readily than neuronal cells. However, the mechanisms by which this occurs in the brain through vitamin D have yet to be fully elucidated, i.e., the mechanisms by which microglia proliferate, are recruited, and change morphology. Possible mechanisms have suggested a role for vitamin D as an anti-oxidative agent and attenuator of neuroinflammation through direct action on microglia [50,142,145]. However, this microglial recruitment and activation could be mediated through neuronal ROS accumulation due to vitamin D deficiency rather than through direct action on the microglia. Considering the interplay between glial cells, neurons, and the extracellular matrix and how vitamin D affects all of these components of the tetrapartite synapse [34,133], it becomes difficult to parse.

While our analyses did not reveal any significant changes in the extracted features of astrocytes, microglial changes were apparent in the *female mice only*. Consistent with previous studies finding an effect of vitamin D on glial cells, vitamin D led to a difference on measures of perimeter, integrated density, and average size and can be considered as proxy measures of phenotype. Although determining pro- or anti-inflammatory states and phenotypes based solely on general extracted features of morphology and count across the DG is not possible (rather than an extensive mapping of a large selection of microglial branching, complexity, and soma size combined with more differentiating surface markers) [146], this does indicate that there is a change in microglial cells in the DG of female mice due to supplementation, a finding echoed in other studies under different conditions [50,134,142]. Microglia have been shown to play a role in the processes of the brain, from learning and memory [134,147], to their effect on lineage differentiation [108,144], and extra-cellular matrices and peri-neuronal nets [147].

Why this effect was sex-specific is worth investigating. Sex-specific attentional deficits have been found for BALB/c mice, for example [66]. More relevant, however, is the evidence that vitamin D deficiency increased the number of activated microglia in the spinal cord and dystrophic microglia in the hippocampus of female but not male mice [142]. Despite these sex-specific differences, the authors were not able to find a significant effect of sex or sex X treatment [142]. Considering VDR is a secosteroid that has been found to have an additive effect with progesterone on astrocytic cells [136] and the extensive literature on its effect on immune cells [8,142,143,148], it is essential to include female rodents and subjects in future studies investigating vitamin D and microglia in the brain.

### 4.4. Future Perspectives

Could vitamin D have opposing effects on proliferation and differentiation depending on geography, i.e., the internal milieu within the brain? How does vitamin D differentially affect the proliferation and differentiation of cells depending on the neurogenic niche? There are still many functions of vitamin D that have yet to be elucidated. This is in addition to many studies linking vitamin D to the development and survival of dopaminergic cells and systems [110,114,123,127,149,150,151]. Moreover, both 1,25(OH)D and 25(OH)D were recently found to be partial agonists of the transient receptor potential vanilloid 1 channel (TRPV1) in the trigeminal neuron by reducing T-cell activation and calcium signaling in the neuron. Interestingly, TRPV1 is a receptor known to play a role in pain, and its agonists are capsaicin and oleoyl dopamine [152,153]. There are still many questions regarding vitamin D’s role in the brain that we are only beginning to explore. Kim, et al. [154], for example, found that vitamin D influenced brain endothelial p-glycoprotein (MDR1a) levels by reversing the effects of induced parkinsonism in mice. P-glycoprotein is an important player/component in the blood–brain barrier, where it pumps out of cells a wide array of substrates. This is further corroborated by other studies that also implicated downstream VDR target MDR1 [155]. Other studies have seen a beneficial effect in models of ischemia or traumatic brain injury by manipulating another vitamin D receptor, Protein Disulfide Isomerase A3 (PDIA3, Erp57) [156,157]. PDIA3 has been linked to amyotrophic lateral sclerosis [158,159], and its deletion is embryonically lethal [160]. These mechanisms deserve further exploration within a more controlled experimental paradigm while taking into consideration the high levels of vitamin D in commercially available rodent chow, the different effects on strains and therefore also humans [161], and the effect vitamin D probably has on all the components of the tetrapartite synapse.

## 5. Conclusions

Our data demonstrated that giving a vitamin D deficient or supplemented diet in adult C57Bl/6 mice over 6 weeks did not affect measures of proliferation or neurogenesis in the neurogenic niches of male or female mice. Moreover, there was no effect of manipulating vitamin D in the diet on astrocyte morphology or count, or on anxiety-like behavior, locomotion, or hippocampal-dependent memory as measured by behavioral tests. There was a sex-specific effect of diet on microglial morphology in female mice, but not male mice. 

## Figures and Tables

**Figure 1 nutrients-16-02938-f001:**
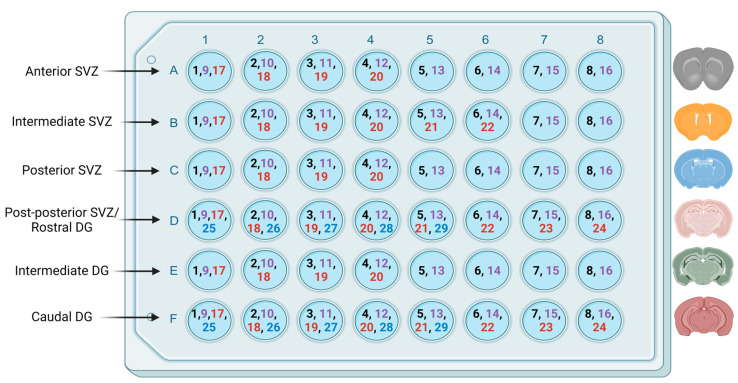
A schematic diagram of the fractionator method as applied to the mouse SVZ and DG. Free-floating coronal sections are distributed in a 48-well plate. According to the mouse atlas, the subventricular zone was subdivided into anterior, intermediate, posterior, and post-posterior regions [82,83,85]. The hippocampus was subdivided into rostral, intermediate, and caudal regions [82]. In each subdivision, the first section was placed in the first well, and the following sections were placed serially in the adjacent wells, reaching the 8th slice in the 8th well (this distribution is color-coded in the figure). The 9th section (tagged by a different color in the figure) was placed in the first well along with the 1st section so that each slice is 280 μm apart from the next slice in the same well/set. Created with BioRender.

**Figure 2 nutrients-16-02938-f002:**
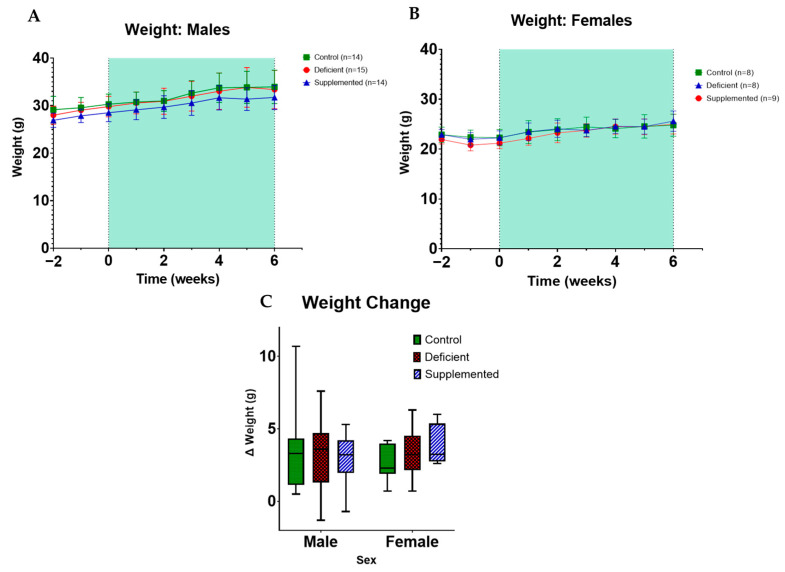
The weights of mice were monitored and recorded weekly. (**A**) Weights of male (n = 43) and (**B**) female mice (n = 25) starting are shown starting 2 weeks before the experimental diet and through the protocol. Weight increased similarly across groups over time (the shaded areas represent the period the mice were on their respective experimental diets). Data are expressed as mean ± SEM. (**C**) There was no difference between groups on weight change from the beginning of the experimental protocol to the end. Data are expressed as mean ± min to max.

**Figure 3 nutrients-16-02938-f003:**
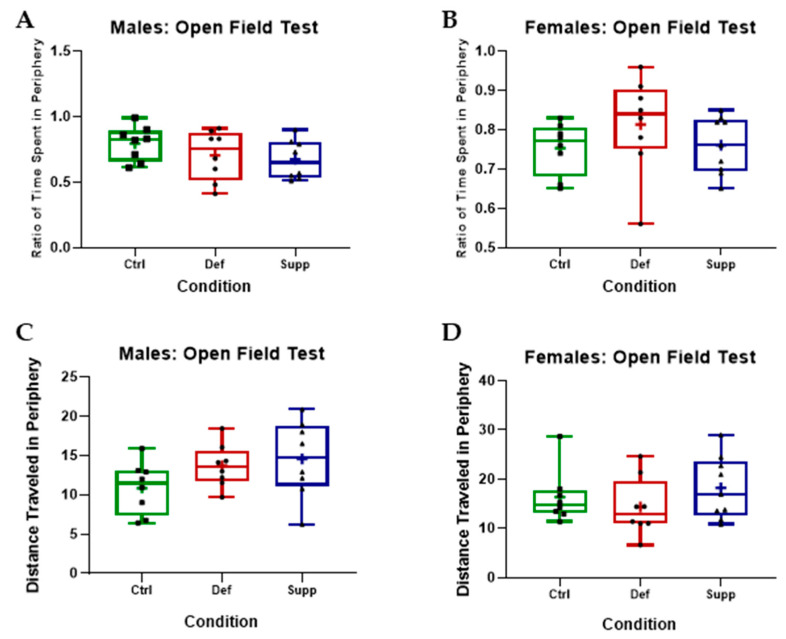
The open field test (OFT) assesses anxiety-like behavior and locomotion. One-way ANOVAs revealed no statistically significant differences between different diets on measures of interest relating to anxiety-like behavior in (**A**) males or (**B**) females. There were also no significant differences between conditions on measures of locomotion in (**C**) males or (**D**) females. Data are presented as box-and-whisker plots from min to max, with all data points. ‘+’ = mean. N = 8–9 per group. Ctrl: control group; Def: deficient group; Supp: supplemented group.

**Figure 4 nutrients-16-02938-f004:**
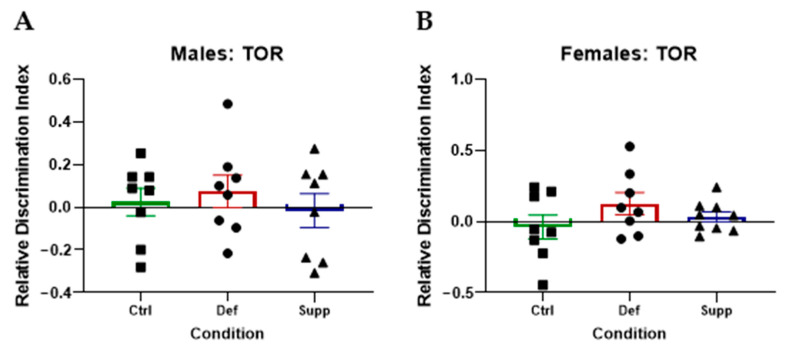
The temporal order recognition (TOR) test was used to assess hippocampal−dependent memory. The relative discrimination index, preference for the more recent object, did not show any significant difference between groups for (**A**) males or (**B**) females (*p* > 0.05). N = 8–9 per group. Data are presented as mean ± SEM. Ctrl: control group; Def: deficient group; Supp: supplemented group.

**Figure 5 nutrients-16-02938-f005:**
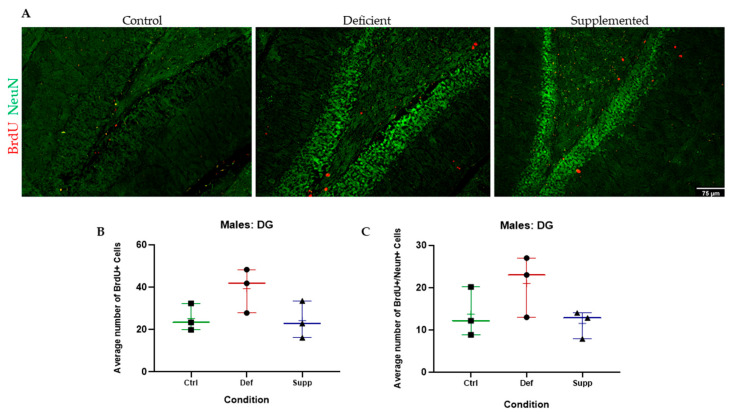
BrdU+ and double BrdU+/NeuN+ proliferating cells were manually counted in the DG of male C57Bl/6 mice. (**A**) Representative images at 20× of the DG of male mice given control, deficient, or supplemented vitamin D diets. There were no significant differences in the number of (**B**) proliferating cells or (**C**) proliferating neurons. N = 3 per group. Data are presented as whisker plots from min to max, with all data points. ‘+’ = mean. Scale bar = 75 μm. Ctrl: control group; Def: deficient group; Supp: supplemented group.

**Figure 6 nutrients-16-02938-f006:**
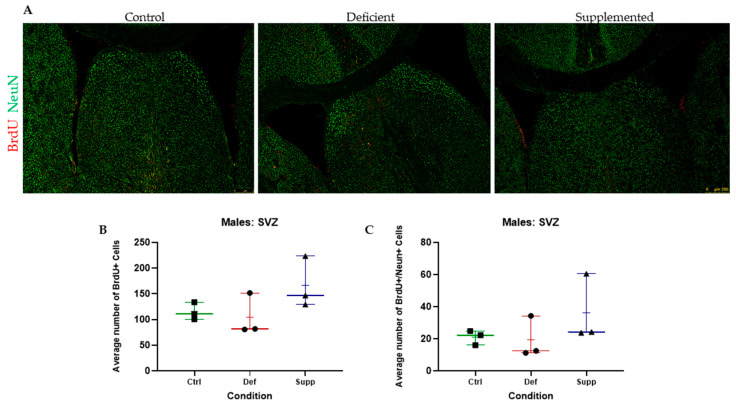
BrdU+ and double BrdU+/NeuN+ proliferating cells were manually counted in the SVZ of male C57Bl/6 mice. (**A**) Representative images at 5× of the SVZ of male mice given control, deficient, or supplemented vitamin D diets. There were no significant differences in the number of (**B**) proliferating cells or (**C**) proliferating neurons. N = 3 per group. Data are presented as mean ± SEM. Scale bar = 250 μm. Ctrl: control group; Def: deficient group; Supp: supplemented group.

**Figure 7 nutrients-16-02938-f007:**
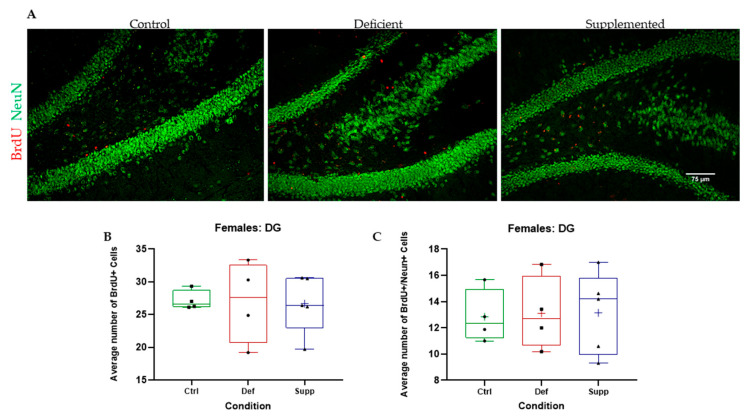
BrdU+ and double BrdU+/NeuN+ proliferating cells were manually counted in the DG of female C57Bl/6 mice. (**A**) Representative images at 20× of the DG of female mice given control, deficient, or supplemented vitamin D diets. There were no significant differences in the number of (**B**) proliferating cells or (**C**) proliferating neurons. N = 4–5 per group. Data are presented as box-and-whisker plots from min to max, with all data points. ‘+’ = mean. Scale bar = 75 μm. Ctrl: control group; Def: deficient group; Supp: supplemented group.

**Figure 8 nutrients-16-02938-f008:**
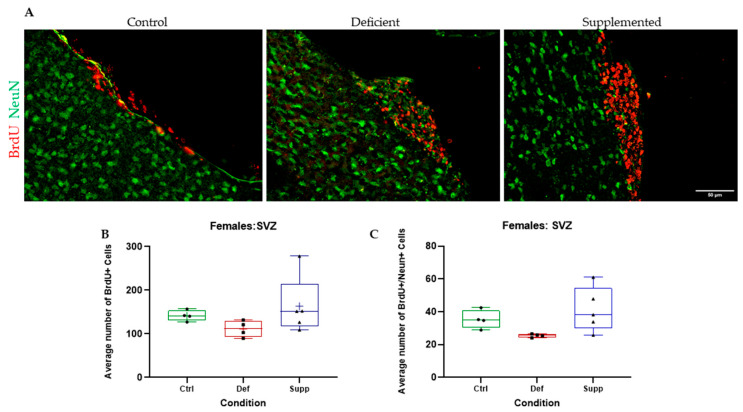
BrdU+ and double BrdU+/NeuN+ proliferating cells were manually counted in the SVZ of female C57Bl/6 mice. (**A**) Representative images at 40× of the SVZ of female mice given control, deficient, or supplemented vitamin D diets. There were no significant differences in the number of (**B**) proliferating cells or (**C**) proliferating neurons. N = 4–5 per group. Data are presented as box-and-whisker plots from min to max, with all data points. ‘+’ = mean. Scale bar = 50 μm. Ctrl: control group; Def: deficient group; Supp: supplemented group.

**Figure 9 nutrients-16-02938-f009:**
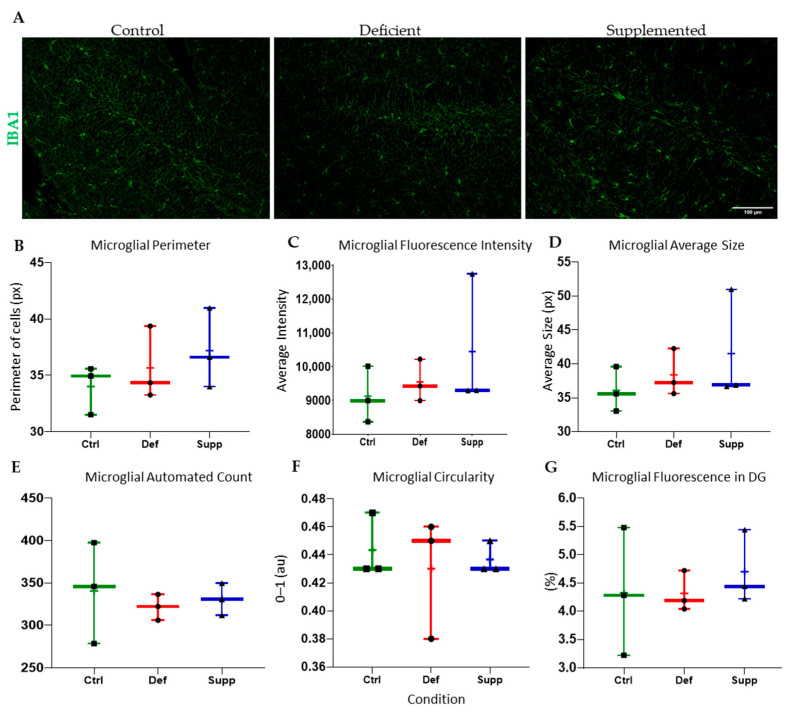
Analysis of microglial features and count in the DG of male mice between diet groups. (**A**) Representative images at 20× of microglia in the DG of male mice given control, deficient, or supplemented vitamin D diets. One-way ANOVAs and post-hoc tests did not reveal any significant difference in extracted features and measures of interest that included microglial (**B**) perimeter, (**C**) fluorescence intensity (Integrated Density), (**D**) average size, (**E**) microglial count, (**F**) circularity, or (**G**) percentage of pixels in the image selection that fluoresced. Data are presented as whisker plots from min to max, with all data points. ‘+’ = mean. N = 3 per group. Scale bar = 100 μm. Ctrl: control group; Def: deficient group; Supp: supplemented group.

**Figure 10 nutrients-16-02938-f010:**
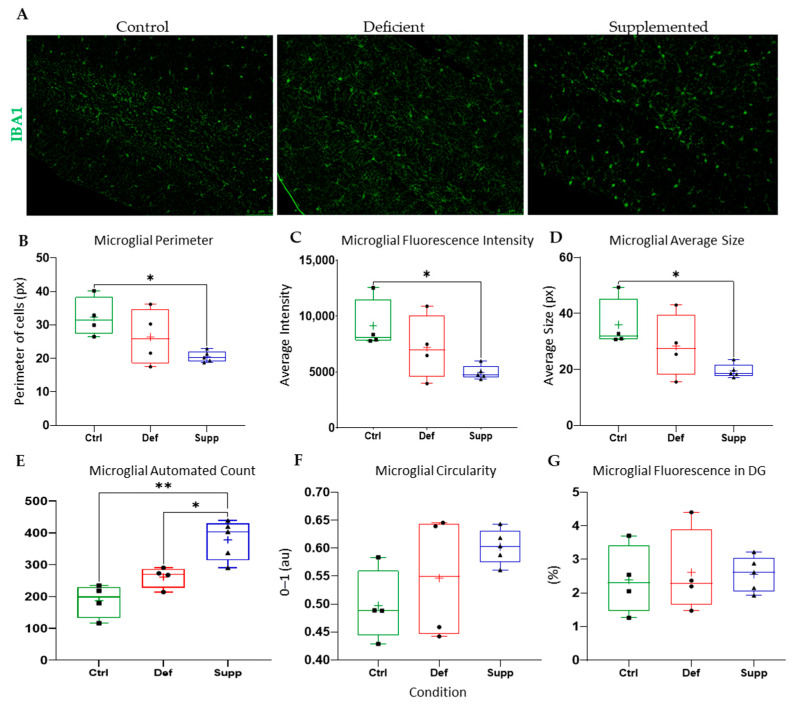
Analysis of microglial features and count in the DG of female mice between diet groups. (**A**) Representative images at 20× of microglia in the DG of female mice given control, deficient, or supplemented vitamin D diets. Extracted features and measures revealed that the VDCG scored significantly higher on (**B**) microglial perimeter, (**C**) microglial fluorescence intensity (integrated density), and (**D**) microglial average size as compared to the VDSG (*p* < 0.05). On the other hand, the VDSG (**E**) microglial count was higher than both the VDDG (*p* < 0.05) and the VDCG (*p* < 0.01). There was no difference between the groups on (**F**) circularity or (**G**) percentage of pixels in the image that fluoresced. n = 4–5 per group. Data are presented as box-and-whisker plots from min to max, with all data points. ‘+’ = mean. *p*-values were derived using post-hoc multiple comparisons and are displayed for datasets encompassing the 3 conditions. * *p* < 0.05, ** *p* < 0.01. Scale bars = 75 μm. Ctrl: control group; Def: deficient group; Supp: supplemented group.

**Figure 11 nutrients-16-02938-f011:**
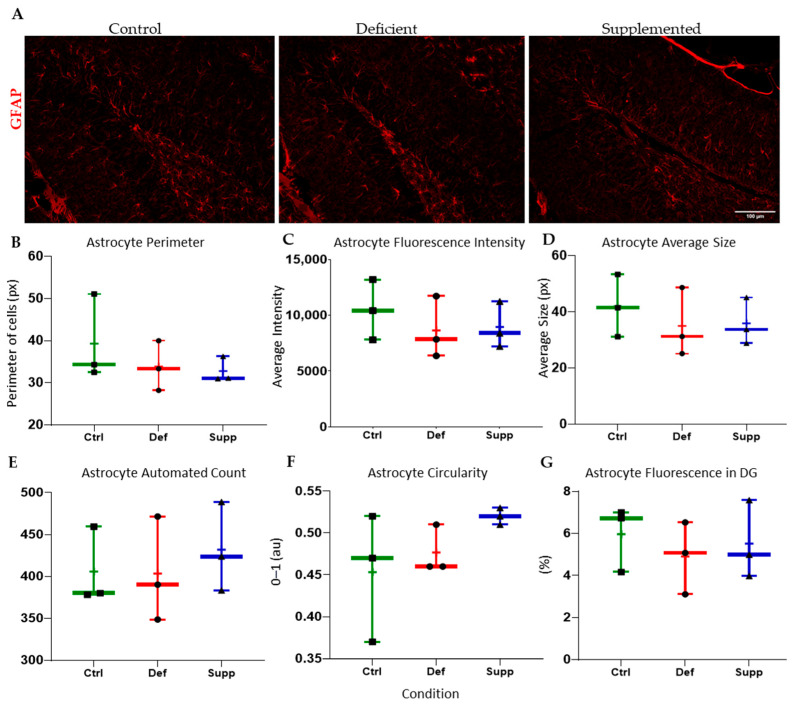
Analysis of astrocyte features and count in the DG of male mice between diet groups. (**A**) Representative images at 20× of astrocytes in the DG of male mice given control, deficient, and supplemented vitamin D diets. One-way ANOVAs and post-hoc tests did not reveal any significant difference in the extracted features and measures of interest that included astrocyte (**B**) perimeter, (**C**) fluorescence intensity (integrated density), (**D**) average size, (**E**) astrocyte count, (**F**) circularity, or (**G**) percentage of pixels in the image selection that fluoresced. Data are presented as whisker plots from min to max, with all data points. ‘+’ = mean. Scale bar = 100 μm. Ctrl: control group; Def: deficient group; Supp: supplemented group.

**Figure 12 nutrients-16-02938-f012:**
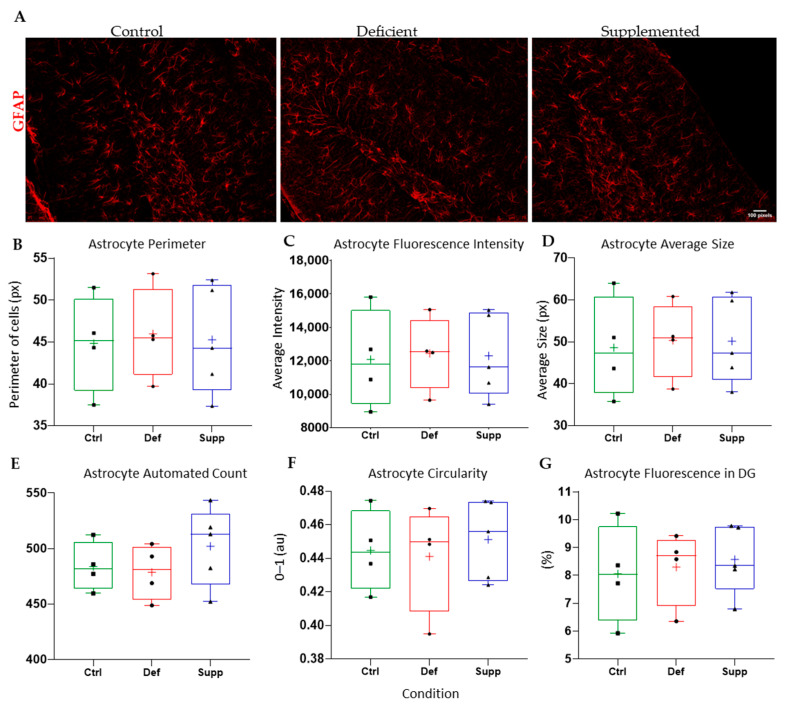
Analysis of astrocyte features and count in the DG of female mice between diet groups. (**A**) Representative images at 20× of astrocytes in the DG of female mice given control, deficient, and supplemented vitamin D diets. One-way ANOVAs and post-hoc tests did not reveal any significant difference in the extracted features and measures of interest that included astrocyte (**B**) perimeter, (**C**) fluorescence intensity (integrated density), (**D**) average size, (**E**) astrocyte count, (**F**) circularity, or (**G**) percentage of pixels in the image selection that fluoresced. Data are presented as box-and-whisker plots from min to max, with all data points. ‘+’ = mean. N = 4–5 per group. Scale bar = 100 μm. Ctrl: control group; Def: deficient group; Supp: supplemented group.

## Data Availability

Datasets are available on request from the authors.

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
