# Peer review of "Vitamin D Deficiency Does Not Affect Cognition and Neurogenesis in Adult C57Bl/6 Mice"

_nutrients, 2024, doi:10.3390/nu16172938_

Round 1

Reviewer 1 Report

Comments and Suggestions for Authors

Line 449 - 450. I could not see where AVD and DVD had been previously defined. Please define.

Comments on the Quality of English Language

English use and convention is adequate.

Author Response

Comments 1: 

Line 449 - 450. I could not see where AVD and DVD had been previously defined. Please define.

Response 1: Thank you for pointing this out. The AVD and DVD definitions have been made clearer in lines:

Lines 425-426

Reviewer 2 Report

Comments and Suggestions for Authors

The study conducted by Doumit et al. evaluated the effect of vitamin D supplementation and deficiency on proliferation and neurogenesis in the main neurogenic niches of the brain, with a focus on microglia and astrocytes. The study is unprecedented and well-conducted, but requires some adjustments for its acceptance:

It is important to present the completed ARRIVE form, in accordance with Nutrients standards;

It would be important to use a group that received only the vitamin D vehicle;

In the item “Vitamin D (25(OH)D) serum measurement” it is important to describe how and if the animals were previously anesthetized to perform this procedure;

What was the reference used for the BrdU dose? It is important to present;

In the sentence “mice underwent cardiac perfusion with 125 4% paraformaldehyde (PFA) after receiving anesthesia”. Again it is important to describe the anesthesia procedure;

I believe that figure 1 is not necessary. Furthermore, it is not described in the body of the method text, only in the caption;

In each behavioral test it is important for the authors to describe the experimental protocol; that is, in what period they were performed after the administration of vitamin D;

Regarding Figure 10, it is important for the authors to describe the methodology of each parameter evaluated from A to G. Furthermore, if the authors add a zoomed-in image of the microglia, using a smaller objective (4X or 10x), it will be more interpretable;

I believe that an experiment with Western blotting could reinforce the findings with microglia (IBa1) in female mice;

Are the effects of vitamin D directly on the microglia or on neurons with consequent release of substances or neurotransmitters that act on the microglia? It is important for the authors to discuss this issue.

Reviewer 3 Report

Comments and Suggestions for Authors

Review nutrients-3146369 12 agosto 2024

The manuscript nutrients-3146369 entitled Vitamin D Deficiency Does Not Affect Cognition and Neurogenesis in Adult C57Bl/6 Mice by Hala Darwish and collaborators, investigated which processes adult vitamin D deficiency affects. , we used. The effect of the vitamin D diets on proliferation in the neurogenic niches, changes in glial cells, as well as on memory, locomotion, and anxiety-like behavior were investigated in vitamin D deficient, control, and supplemented diets over 6 weeks in male and female C57Bl/6 mice

Vitamin D deficiency and supplementation did not affect proliferation, neurogenesis, or astrocyte changes, and this was reflected on behavioral measures. Supplementation only affected microglia in the dentate gyrus of female mice. Indicating that vitamin D deficiency and supplementation do not affect these processes over a 6-week period.

Thge work is scientifically sounding.

Methods for vitamin D quantification are suboptimal and this should be discussed further (see below).

There are several interesting findings about the effect of vitamin D deficiency on the adult mouse brain.

Figures are informative but the authors are recommended to modify these (see below).

Discussion is consistent with results.

English is good

References are appropriated.

Line 23: To investigate which processes adult vitamin D deficiency affects, we used vitamin D deficient, control, and supplemented diets over 6 weeks in male and female C57Bl/6 mice.

To investigate the processes affected by vitamin D deficiency in adults, we studied vitamin D deficient, control, and supplemented diets over 6 weeks in male and female C57Bl/6 mice.

Line 90: there is the Protocol Approval (Approval #17-05-414)

Line 113: The serum was analyzed via a Roche Cobas 8000 (e801) - an electrochemiluminescence binding 114 assay measuring 3.0 - 120 ng/ml (or 7.5 – 300 nmol/L) to confirm vitamin D levels.

The Roche ECLIA method was standardised for humans, not for mice.

Moreover every ECLIA method is far form accuracy and specificity being prone to several interferences.

HLPC-MS/MS is the best methodology for such experimental work.

This should be discussed in the discussion evidencing the suboptimal methodology and discussing the lack of accuracy of ECLIA methods.

Line 281: instead of a box and dot plot, the figure 4 should be presented as a whiskers box plot + the dots of the quantification.

Or just as a dot plot without the box which is inappropriate.

Figures 6, 7, 8, 9, 10, 11 and 12 should be presented as whiskers-box plot + the dots of the results or just with the dots and the error bars.

Box plot as presented is misleading and inappropriate.

Line 457: the authors did not find differences in OFT and TOR which is different by other reports.

This finding should be more deeply discussed.

Comments on the Quality of English Language

english is good

Round 2

Reviewer 3 Report

Comments and Suggestions for Authors

The authors answered to the comments and improved the manuscript